# Development of an MRI-Guided Approach to Selective Internal Radiation Therapy Using Holmium-166 Microspheres

**DOI:** 10.3390/cancers13215462

**Published:** 2021-10-30

**Authors:** Joey Roosen, Mark J. Arntz, Marcel J. R. Janssen, Sytse F. de Jong, Jurgen J. Fütterer, Christiaan G. Overduin, J. Frank W. Nijsen

**Affiliations:** 1Department of Medical Imaging, Radboud Institute for Health Sciences, Radboud University Medical Center, 6525 GA Nijmegen, The Netherlands; mark.arntz@radboudumc.nl (M.J.A.); marcel.janssen@radboudumc.nl (M.J.R.J.); jurgen.futterer@radboudumc.nl (J.J.F.); kristian.overduin@radboudumc.nl (C.G.O.); frank.nijsen@radboudumc.nl (J.F.W.N.); 2Department of Cardiothoracic Surgery, Radboud University Medical Center, 6525 GA Nijmegen, The Netherlands; sytse.dejong@radboudumc.nl

**Keywords:** SIRT, radioembolization, holmium, MRI, safety, compatible, conditional, angiography, catheter

## Abstract

**Simple Summary:**

Selective internal radiation therapy (SIRT) is a treatment for patients with liver cancer that involves the injection of radioactive microspheres into the liver artery. For a successful treatment, it is important that tumours are adequately covered with these microspheres; however, there is currently no method to assess this intraoperatively. As holmium microspheres are paramagnetic, MRI can be used to visualize the holmium deposition directly after administration, and possibly to adapt the treatment if necessary. In order to exploit this advantage and provide a personally optimized approach to SIRT, the administration could ideally be performed within a clinical MRI scanner. It is, however, unclear whether all materials (catheters, administration device) used during the procedure are safe for use in the MRI suite. Additionally, we explore the capability of MRI to visualize the microspheres in near real-time during injection, which would be a requirement for successful MRI-guided treatment. We further illustrate our findings with an initial patient case.

**Abstract:**

Selective internal radiation therapy (SIRT) is a treatment modality for liver tumours during which radioactive microspheres are injected into the hepatic arterial tree. Holmium-166 (^166^Ho) microspheres used for SIRT can be visualized and quantified with MRI, potentially allowing for MRI guidance during SIRT. The purpose of this study was to investigate the MRI compatibility of two angiography catheters and a microcatheter typically used for SIRT, and to explore the detectability of ^166^Ho microspheres in a flow phantom using near real-time MRI. MR safety tests were performed at a 3 T MRI system according to American Society for Testing of Materials standard test methods. To assess the near real-time detectability of ^166^Ho microspheres, a flow phantom was placed in the MRI bore and perfused using a peristaltic pump, simulating the flow in the hepatic artery. Dynamic MR imaging was performed using a 2D FLASH sequence during injection of different concentrations of ^166^Ho microspheres. In the safety assessment, no significant heating (ΔT_max_ 0.7 °C) was found in any catheter, and no magnetic interaction was found in two out of three of the used catheters. Near real-time MRI visualization of ^166^Ho microsphere administration was feasible and depended on holmium concentration and vascular flow speed. Finally, we demonstrate preliminary imaging examples on the in vivo catheter visibility and near real-time imaging during ^166^Ho microsphere administration in an initial patient case treated with SIRT in a clinical 3 T MRI. These results support additional research to establish the feasibility and safety of this procedure in vivo and enable the further development of a personalized MRI-guided approach to SIRT.

## 1. Introduction

In interventional oncology, selective internal radiation therapy (SIRT) has become an established treatment modality for primary and secondary liver malignancies [1,2]. During treatment, radioactive microspheres containing the beta-emitters yttrium-90 (^90^Y) or holmium-166 (^166^Ho) are injected into the hepatic arterial tree and transported through the perfused liver volume until they get stuck in the arterioles because of their size. Both isotopes are high-energy β-emitters (^90^Y: E_β-max_ = 2.28 MeV (100%), ^166^Ho: E_β-max_ = 1.85 MeV (48.8%), 1.77 MeV (49.9%)) and their half-lives are 64.2 (^90^Y) and 26.8 h (^166^Ho) [3,4]. As the range of the deposited beta energy is limited to the millimetre range, the microspheres represent an excellent carrier for microbrachytherapy of tumours that are difficult to treat through more conventional treatment regimens such as chemotherapy or external beam radiation therapy (e.g., hepatocellular carcinoma). Clinical studies investigating the efficacy of SIRT using either isotope have resulted in similar clinical outcomes [5].

Even though SIRT has been used in clinical practice for over 20 years, the exact mechanisms behind the distribution of microspheres remain a ‘black box’, and this complicates the prediction of dose distribution and response. In the current clinical setting, a simulation of treatment is performed by injecting technetium-99m labelled macroaggregated albumin (^99m^Tc-MAA) [6] or a very low dose of ^166^Ho microspheres [7] and activity prescription is (partly) based on these simulations. Imaging to visualize the actual dose distribution is typically performed through ^90^Y-PET/CT [8] or ^166^Ho-SPECT/CT [9,10] in the hours to days after treatment.

In a recent review on MRI-guided external radiation therapy for liver tumours, it was shown that image-guided treatment personalization could lead to a reduction in radiation-induced toxicity, while increasing the tumour dose [11]. A similar benefit of personalizing the radiation dose was demonstrated in SIRT in the DOSISPHERE-01 trial, leading to both an increase in local response (71% vs. 36%) and survival (26.6 mo vs. 7.1 mo), without an increase in adverse events [12]. Moreover, a lot of research has investigated the dose–response relation in SIRT and it has been shown that an improved tumour dose leads to an improved response, both in terms of liver progression-free and in overall survival [13].

Next to SPECT/CT, ^166^Ho microspheres can also be visualized and quantified through MRI, as holmium is a paramagnetic metal (which is not possible using yttrium-based microspheres). Acquired images can then be translated to MRI-based dose maps fit for dosimetry [14,15,16]. Subsequently, if the microspheres were to be administered while the patient is positioned in an MRI scanner, the physician could assess the expected tumour dose on a voxel level in semi-real time and adjust the treatment parameters if deemed necessary. This way, SIRT may become a truly adaptive, image-guided procedure.

In preparation for a clinical trial designed to investigate the feasibility of an MRI-guided approach to SIRT in salvage patients [17], we addressed several hurdles that needed to be overcome to allow SIRT to be delivered in a clinical MRI environment. Firstly, MRI-guided catheter-based vascular interventions are scarce and, to the best of our knowledge, no clinically available angiography catheters and guidewires are registered as MR-conditional for use at 3 Tesla (T) [18]. Additionally, previous studies into imaging ^166^Ho microspheres with MRI [14,15,16] and SPECT [19,20] have mainly focused on the quantification once the microspheres are already lodged in the liver arterioles, i.e., in their final position. Ideally, MRI guidance would also be used to visualise the loss of signal caused by the microspheres in near real-time during administration, in order to be able to terminate the administration of microspheres if stasis or backflow occur. Although some previous studies have demonstrated near real-time visualization of holmium microsphere injection under MRI in rabbits [21] and a porcine model [22], its dependency on factors such as holmium concentration and flow are unknown.

Therefore, the aim of the current study was to investigate the MR safety of three conventional angiography catheters for use during MRI-guided SIRT at 3 T in an ex vivo setting and to investigate the correlation between ^166^Ho microsphere concentration and detectability in near real-time MRI in a flow phantom. Finally, we correlate our findings with respect to catheter visibility and near real-time imaging of the microspheres in an initial patient case.

## 2. Materials and Methods

All experiments were performed using a 3 T clinical MRI scanner (MAGNETOM Skyra, Siemens Healthineers, Erlangen, Germany). A spine and body phased-array coil were used for all imaging.

### 2.1. MR Safety

The following materials were assessed for MR safety: an 80 cm, 5 Fr, C1-shaped angiography guiding catheter (Radifocus Glidecath, Terumo, Tokyo, Japan), referred to as catheter A; a 65 cm, 4 Fr, C1-shaped angiography guiding catheter (Tempo, Cordis, Santa Clara, CA, USA), referred to as catheter B; and a 150 cm, 2.8 Fr microcatheter (Progreat, Terumo, Tokyo, Japan), referred to as microcatheter. As the microcatheter is always inserted within a guiding catheter, it was also tested for MR safety in that configuration, and MR safety tests were performed following the American Society for Testing of Materials (ASTM) standard test methods [23,24,25], which are guidelines on MRI safety testing with respect to RF-induced heating, the magnetic displacement force and image artefact size. All materials were embedded in a tissue-mimicking agarose phantom (74 × 32 × 9 cm), of which a schematic overview is drawn in Figure 1 (phantom design with catheter B is reported in Appendix A). The phantoms consisted of 30 g of agarose (Merck-Millipore, Burlington, NJ, USA) and 2 g NaCl per litre of demineralized water in order to obtain tissue equivalent dielectric properties. Prior to casting of the phantoms, three fibre optic temperature probes (Neoptix Inc, Quebec, QC, Canada) were attached to the investigated material at three reference points. In both guiding catheters (Figure 1A), these were located at the catheter tip, mid-way in the C-curve and more proximal along the catheter shaft. In the phantom with the microcatheter, these were located at the tip, at a marker 3 cm proximal from the tip, and along the guiding catheter shaft (Figure 1B). Additionally, one temperature probe was placed on the side of the phantom as a reference.

#### 2.1.1. Radiofrequency (RF)-Induced Heating

The phantoms were placed as close to the edge of the bore as possible, in order to simulate as extreme conditions as possible. As a direct result of the length of the microcatheter/guiding catheter combination, the guiding catheter was placed even closer (closest distance to the edge of the phantom approximately 5 cm as opposed to 15 cm) to the MRI bore in the phantom with the microcatheter inserted (Figure 1B). MRI was performed using four clinical routine pulse sequences (T1 VIBE, T2 TSE, T2 HASTE, TRUFI, see Table 1) and one sequence that had been modified to produce the maximum allowed RF-power (TRUFI RF_max_) in order to maximize the specific absorption rate (SAR). A detailed overview of investigated sequence types and corresponding parameters is displayed in Table 1. In order to quantify the time-averaged RF-power and whole-body SAR, a healthy volunteer was imaged for 6 min per sequence on first-level mode (maximum whole-body SAR of 4 W/kg). The whole-body SAR as expressed by the MR scanner console directly after the sequence was completed is reported.

Each sequence was acquired in triplo for 1.5 min, representing a realistic imaging time during MRI-guided interventions. In order to vary the insertion depth of the catheters, a 5 cm-wide block of agarose was removed twice (see Figure 1), after which all measurements were repeated. This resulted in insertion depths of 60, 65 and 70 cm (catheter A), 55, 60 and 65 cm (catheter B) and 70, 75 and 80 cm (microcatheter). Throughout the experiments, temperature recordings were captured every second. Maximum temperature rise during imaging was calculated and corrected for any background temperature rise of the whole phantom by subtracting any temperature rise detected at the reference sensor. Afterwards, the phantoms were carefully dissected in order to visually confirm that the temperature probes had not dislocated during casting of the agarose.

#### 2.1.2. Magnetically Induced Displacement and Artefact Size

All abovementioned materials were placed on the MRI table close to the bore, in order to assess displacement due to the magnetic field through visual observation. In a similar agarose phantom as in the RF-induced heating experiments, all materials were embedded in a single phantom in order to assess the maximum artefact size on images acquired through the abovementioned imaging sequences. The maximum artefact size was calculated in MATLAB R2018a (Mathworks, Natick, MA, USA), in which the artefact was defined as voxels with an intensity higher or lower than the mean background intensity ± 3 times the standard deviation of the background intensity.

### 2.2. Near Real-Time MRI Visibility of Microspheres

#### 2.2.1. Flow Phantom

A flow phantom was created by casting a mixture of polyvinyl alcohol (PVA; 7.0 wt%, Sigma-Aldrich, Saint Louis, MO, USA), ethylene glycol (38.0 wt%, Sigma-Aldrich, Saint Louis, MO, USA) and demineralised water around plastic tubing with an outer diameter of 5 mm. The mixture was heated up to 90 °C for one hour prior to casting, until all PVA had dissolved. It was placed in a freezer at −20 °C overnight and kept in a refrigerator at 4 °C until the experiment. A schematic of the phantom and experimental setup is drawn in Figure 2.

Non-irradiated holmium-165 PLLA microspheres (^165^Ho microspheres; Quirem Medical B.V., Deventer, The Netherlands) were suspended in 0.1% Pluronic solution (Quirem Medical B.V., Deventer, The Netherlands) in order to prepare the following concentrations: 10, 25, 50, 75 and 100 mg/mL. The flow phantom was placed in the MRI bore and attached to a peristaltic pump (HL20, Getinge, Gothenburg, Sweden) that was set to perfuse the system continuously with 0.9% NaCl at both 50 and 100 mL/min to simulate the flow in the distal right hepatic artery. The microcatheter was inserted in the lumen through a 5 Fr vascular sheath (BRITE TIP^®^ sheath, Cordis, Santa Clara, CA, USA) that had been glued into a piece of plastic tubing in order to inject the holmium microspheres during MR imaging, similar to the in vivo situation.

To visualize the injection of the holmium microspheres and its flow pattern through the phantom, a near real-time 2D FLASH sequence was used (TE 6.6 ms, TR 10 ms, flip angle 33°, single slice, slice thickness 6 mm, in plane resolution 1.48 × 1.48 mm) with a temporal resolution of 0.8 s. The sequence was initiated and after a couple of images had been captured, 1 mL of holmium microspheres suspension was injected into the microcatheter manually, followed by flushing the microcatheter with 5 mL NaCl 0.9%. Each measurement was performed in triplo.

#### 2.2.2. In Vivo Procedure

As part of an ongoing trial at our institution (EMERITUS study, clinicaltrials.gov identifier NCT04269499), patients were treated with MRI-guided SIRT with holmium-166 microspheres for hepatic malignancies. Local ethics committee approval was obtained (reference number: NL68354.091.18) and all patients provided written informed consent. In this work, we present preliminary data of one included patient to correlate with phantom findings. Full details on the inclusion/exclusion criteria and exact study procedures will be published elsewhere once the study is completed, and are available online at https://clinicaltrials.gov/ct2/show/NCT04269499, accessed on: 10 October 2021).

In brief, on the day of treatment, the guiding catheter and microcatheter were positioned in the hepatic arterial system under X-ray guidance as per usual. Thereafter, the patient was transferred to an adjacent clinical 3 T MRI system available at the operating room (OR) suite. A volumetric T1-weighted, FLASH sequence (TE 3.7 ms, TR 7.8 ms, flip angle 10°, slice thickness 2 mm, in plane resolution 0.78 × 0.78, number of slices: 32) was used to verify the unchanged position of the microcatheter after transfer. Hereafter, a routine dosage of holmium-166 (60 Gy mean liver dose, calculated as in the HEPAR-2 trial [5]) microspheres was administered in four fractions (10/30/30/30%) under near real-time MR imaging using the same single-slice FLASH sequence as in the flow phantom experiment (temporal resolution of 0.8 s). The near real-time imaging slice was selected so as to capture a part of the arterial tree downstream of the catheter tip, and if possible, also include tumour tissue. The patient was asked to hold their breath for as long as possible (approximately 30–40 s), and the administration of microspheres started as soon as the near real-time MR images were visualized on a LED-monitor in the MRI room.

### 2.3. Image Analysis

Image analysis was performed in MATLAB R2018a. In the flow phantom data, 2 volumes of interest (VOIs) were drawn (see Results Section 3.4): one immediately downstream of the catheter tip (4 × 11 voxels), and one further downstream, halfway down the length of the tubing (4 × 19 voxels). In the in vivo data, four different VOIs were drawn (see Results Section 3.5): one proximal in the hepatic artery (29 voxels), one more distal in the hepatic artery (34 voxels) and two VOIs at two tumours (edge of large tumour: 79 voxels, small tumour: 34 voxels). Signal intensities were determined for every voxel within the VOI, data are presented as a mean intensity per VOI.

### 2.4. Statistics

As a result of low sample size and the descriptive character of the investigated subject, no statistical analysis was performed. Data are reported as the arithmetic mean with the range in brackets.

## 3. Results

### 3.1. RF-Induced Heating

The maximum heating found for catheter A and catheter B in routine clinical positions (Figure 1A) was 0.2 °C (range: 0.1–0.2 °C) and 0.1 °C (range: 0.0–0.2 °C), respectively. Insertion of the microcatheter into catheter A resulted in an extreme off-centre position (close to the bore) of catheter A (see Figure 1B), during which a maximum heating of 0.7 °C (range: 0.6–0.7 °C) was found along the catheter shaft at an insertion depth of 65 cm using the TRUFI RF_max_ sequence. Additional heating data of catheter A in this orientation are presented in Table 2. The maximum found heating in the microcatheter was 0.1 °C (range: 0.0–0.3 °C).

In Figure 3, a representative example of the temperature measurements on catheter A during imaging with the T1 VIBE sequence and TRUFI RF_max_ sequence is plotted over time. After the sequence was aborted, the material rapidly cooled down to its starting temperature.

### 3.2. Magnetically Induced Displacement and Artefact Size

Catheter A and the microcatheter did not show any displacement due to magnetic interaction. Catheter B was slowly pulled into the bore by the magnetic field when placed on the MR table closer than 40 cm to the entrance to the bore.

The artefacts induced by the different catheters are visualized in Figure 4, and the maximum artefact diameters are reported in Table 3. In catheter A, a clear difference in artefact size was found between the majority of the catheter length (maximum of 8.1 mm on TSE) and the more flexible, most distal 10 cm of the catheter (maximum of 3.1 mm on FLASH). Catheter B resulted in a larger artefact than catheter A in all three sequences (up to 36.9 mm). The microcatheter had the relatively smallest artefact along the shaft (2.4 mm), with a maximum artefact of 4.7 mm being induced by the platinum/iridium marker at the tip.

### 3.3. In Vivo Artefact Size

Based on these results, catheter A was selected as the guiding catheter during the clinical study. In Figure 5, an example of imaging acquired for localization of the catheters is shown. Catheter A exiting the aorta and entering the common hepatic artery is clearly recognisable; however, the small artefact at the tip of the microcatheter is difficult to detect.

### 3.4. Near Real-Time MRI Visibility of Microspheres

Figure 6 shows an example of near real-time MR images during the injection of 100 mg/mL ^165^Ho microspheres at a flow speed of 50 mL/min. Directly after the initial injection and saline flush, the bolus of holmium microspheres induces a signal drop in the vessel. In Figure 7, the minimum VOI signal intensity is shown as a function of holmium concentration and pump speed. The maximum relative signal loss observed just distal from the catheter tip was 1%, 3%, 6%, 4% and 12% (range: 9–14%) after injection of, respectively, 10, 25, 50, 75, and 100 mg/mL holmium at a pump speed of 50 mL/min. When pump speed was increased to 100 mL/min, this decreased to 3%, 2%, 4%, 6% and 6%, respectively. The relative signal loss downstream was always lower than directly distal from the catheter tip, with a maximum of 5% (range: 4–5%) at 100 mg/mL holmium and a pump speed of 50 mL/min. A linear correlation between the concentration of microspheres and the extent of signal loss was found in all four scenarios, albeit most clear near the catheter tip (R^2^ = 0.84 and 0.91) and at a pump speed of 50 mL/min further downstream (R^2^ = 0.81).

### 3.5. In Vivo Visibility during Holmium-166 SIRT

SIRT was performed in a 74-year-old patient with intrahepatic cholangiocarcinoma. The main lesion was located in Couinaud segments 4A and 4B, and there were multiple satellite lesions in all other liver segments (see Figure 9A,B). A whole-liver SIRT was performed under near real-time MR-imaging, starting with the right hemiliver. The microcatheter was positioned proximal in the right hepatic artery (RHA). In Figure 8, angiography of the entire liver and selective angiography of the RHA are visualized. In Figure 9C–F, different chronological frames of the near real-time imaging during the injection of 30% of the total dose of holmium microspheres are presented. A video of this near real-time imaging series is available online (Appendix A).

The aforementioned holmium microspheres were administered in multiple smaller injections (total amount = 126 mg), with catheter flushing with NaCl 0.9% in between, as per usual when using the Customer Kit for microsphere injection. During the first half of the near real-time imaging, a loss of signal is seen in the proximal RHA (Figure 9D). In the second half, this loss of signal was mainly seen more distally in the RHA (Figure 9E), probably as a result of a small breathing motion of the patient, because of which the proximal RHA moved out of plane and the distal RHA entered the imaging plane. Almost immediately after start of injection, loss of signal is seen accumulating in the tumours (Figure 9F).

Quantification of the signal intensities in four different VOIs (proximal RHA (I), distal RHA (II), the edge of the large tumour (III), and the small tumour (IV)) are visualized in Figure 10. The transient loss of signal during the initial injections of the microspheres were clearly visible in the proximal RHA, with the last drop in intensity around 30–40 s, after which the loss of signal occurred more distally in the RHA. In both tumours, there was a rapid decrease in signal intensity during the first injection of microspheres, after which the intensity kept decreasing during the remainder of the injections.

## 4. Discussion

The goal of this study was to pave the way towards a personalized SIRT procedure under MRI guidance for patients with liver tumours. The first attempts at MR guided SIRT were performed 15 years ago in laboratory animals (pigs) in our research group [21,22]. In the current study, we established the MR safety of two angiography catheters and a microcatheter that are already clinically available and commonly used for the procedure in clinical practice, in an ex vivo setting. Additionally, we investigated and demonstrated the feasibility of near real-time imaging of ^166^Ho microspheres in a flow phantom using a single slice 2D FLASH sequence, and compared our findings to an initial patient case.

Both of the investigated angiography catheters (catheter A and catheter B) contain a metallic braiding that is fully coated with nylon and other polymers. The manufacturer of both catheter A and catheter B only specified ‘stainless steel’ as the used material for their metallic braiding, making it impossible to estimate the MRI-compatibility beforehand [26,27,28]. Interestingly, both were found to be conditionally MRI-compatible for use in SIRT in a 3 T MRI system, despite this metallic braiding. The maximum heating found in catheter A under extreme (not realistic for in vivo use) conditions was 0.7 °C, which is probably an overestimation, as in vivo it will be cooled by the blood and fluids injected through the lumen of the catheter. All RF heating found in other materials was <0.2 °C. Comparing catheter A and B, two disadvantages of catheter B were identified. The first is the increased artefact size in all three investigated sequences (maximum difference of 8.1 vs. 36.9 mm, TSE). This would impede the accurate identification of the vascular structures, such as the hepatic artery with a diameter of approximately 3 mm, during SIRT. The second disadvantage is the magnetic displacement found when catheter B was placed close to the MRI bore. This could potentially alter the catheter position in vivo, leading to dangerous situations (damage to the vascular wall, extrahepatic deposition of radioactivity), and therefore, no measurable interaction (catheter A) is strongly preferred, and catheter B is classified as MR-unsafe. Finally, although we show the acceptable safety of catheter A with a microcatheter under the tested conditions, the translation of the results to the in vivo setting should always be evaluated in the specific situation and assessed with local MR safety authorities before clinical use.

The investigated microcatheter contains a spiral structure composed of tungsten, fully coated with a polymer [29]. Tungsten is paramagnetic and therefore at risk for RF-induced heating, which we, however, did not observe in this study. Both the tungsten braiding and the platinum/iridium markers at the tip and 3 cm proximal from the tip of the microcatheter evoked a very small artefact on MRI, making it feasible but difficult to precisely locate the catheter tip both in and ex vivo. Altering the tip markers may be recommendable to improve detectability under MRI.

In the flow phantom, the extent of local signal loss as a result of the administration and passing of ^165^Ho microspheres through the imaging volume was linearly dependent on injected concentration of microspheres and as a function of the pump speed of the fluid flowing through the tubing. A similar experiment was performed in 2004 [21], albeit using a very different setup. In the cited study, a flow phantom was built to mimic the flow in the inferior vena cava, with the goal of detecting the shunting of microspheres between the hepatic arterial system and the vena cava in near real-time. The used vessel had a larger diameter (12 vs. 5 mm) and the microspheres were injected using a 5 Fr catheter as opposed to the clinically used 2.8 Fr microcatheter. Nonetheless, similar decreases in signal intensity were found.

The main limiting factor in our phantom experiment is the lack of scientific data regarding the flow speed of blood in the hepatic arterial system, especially when a catheter is placed in the artery. Comparing the phantom data to the presented in vivo case, the signal decreased to a greater extent in vivo, which suggests that the flow velocity in this patient was lower than the lowest velocity in the phantom experiment. As the administration of the ^166^Ho microspheres in the clinic are administered using a shielded administration box [30], it is very difficult to calculate the injected concentrations of microspheres. In a recent publication, it was shown that the majority of the microspheres are administered in the first few injections, exponentially decreasing each injection [31]. This also complicates a proper comparison with the flow phantom. Another limitation of the current study is that the in vivo data are based on only a single patient case and, therefore, it has to be noted that the acquired imaging information and/or quality may be patient dependent.

There are still some pitfalls in working towards an MRI-guided SIRT procedure in a hybrid operating room. The first is localization of the catheter tip, especially of the microcatheter: given the small artefact caused by the current platinum/iridium marker, it is hardly recognisable in vivo. In the study by Seppenwoolde et al., a dysprosium marker was attached to the catheter tip, making it clearly visible in pigs [22]. Catheters with such markers are, however, currently (to the best of our knowledge) not clinically available. Another difficulty during near real-time imaging is the visualisation of the blood vessel during injection of microspheres, as in our protocol, only a single slice was imaged during injection. In the presented case, it was demonstrated that only a small breathing motion can already move the targeted blood vessel out of plane. This could potentially be resolved with improved catheter visibility, as the catheter could be a surrogate for locating the blood vessel. Addition of a marker better fit for passive tracking (e.g., ferrous oxide, dysprosium oxide) could greatly enhance the visibility of the used microcatheter [32]. Moreover, an MRI-guided procedure would greatly benefit from continuous 3D imaging during the injection. This could resolve the issue of the vessel moving out of plane and would further increase the safety of the procedure, as it allows for continuous visualisation of the microsphere injection and catheter position, but further work is needed to investigate the feasibility and implementation of such acquisition methods. Another pitfall is the repositioning of the (micro)catheter if there is an inadequate distribution of microspheres. At present, repositioning requires one to go back to the angio suite. Improved visibility of the microcatheter, e.g., using active tracking or specifically designed passive markers, could potentially allow in-bore catheter manipulation, analogous to MRI-guided cardiac interventions.

A fully (cone-beam) CT-guided procedure could present an alternative to MRI-guided SIRT, especially as the catheters are clearly visible using CT-imaging. Feasibility of CT-based quantification of holmium microspheres has been demonstrated multiple times, albeit not after SIRT but after direct intratumoural injection, in rabbit tumour models [33,34] and patients with head-and-neck cancer [35]. Local concentrations of holmium are speculated to be higher after intratumoural injection than after SIRT, which would hamper CT-quantification after SIRT, but this could probably be resolved using holmium microspheres with a higher concentration of holmium per sphere, such as holmium hydroxide or holmium phosphate microspheres [36,37]. An upside of using a CT-guided approach is the possibility to reposition the (micro)catheter under angiographic guidance, for which an MRI-guided approach cannot be used. An intrinsic upside of MRI-guidance is, however, the improved soft-tissue contrast compared to CT, possibly allowing for more accurate delineation of target VOIs and subsequently, more accurate dosimetry.

## 5. Conclusions

In this study, no significant heating or magnetic interaction was demonstrated for one commonly used angiography catheter and a microcatheter under the tested ex vivo circumstances at 3 Tesla. Near real-time MRI visualization of holmium microspheres during administration was feasible in a flow phantom with holmium-induced signal loss increasing linearly with holmium concentration and decreasing at higher vascular flow speeds. We have correlated our findings based on the visualization of the catheters and near real-time imaging of the microspheres in an initial patient case, using the developed workflow for MRI-guided administration of holmium microspheres. Additional research is needed to confirm procedural feasibility and safety in vivo and an ongoing clinical trial has been designed to this purpose, of which the results are awaited. When proven safe, the ability to administer microspheres within the MRI, combined with MRI-based microsphere quantification, would provide new opportunities for a truly adaptive and personalised SIRT procedure.

## Figures and Tables

**Figure 1 cancers-13-05462-f001:**
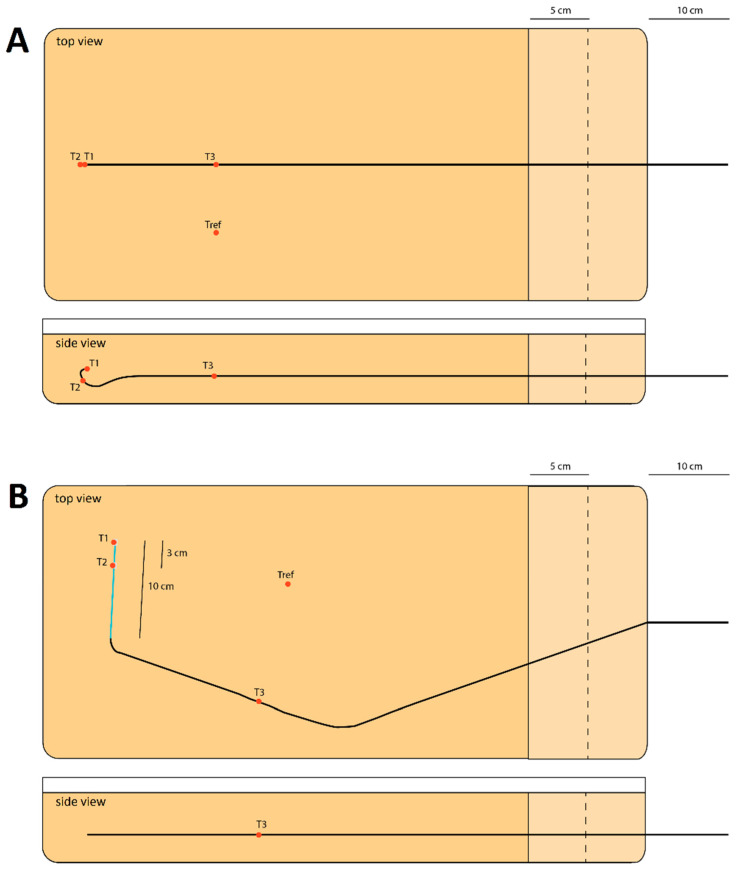
A schematic overview of the agarose phantoms that were cast to investigate the MR safety in guiding catheter A (**A**) and the microcatheter inserted in guiding catheter A (**B**). The black line resembles guiding catheter A, the blue line resembles the microcatheter, and the red dots resemble the temperature probes. In (**A**), probe 1 (T1) is located at the tip of the catheter, probe 2 (T2) midway in the C-shaped curve and probe 3 (T3) along the straight, more proximal part. In (**B**), T1 is located at the tip of the microcatheter, T2 at a marker 3 cm proximal from the tip and T3 along the straight part of guiding catheter A. Tref is a reference temperature probe. After all initial measurements, a 5 cm-wide block of agarose was removed twice near the proximal end of the catheter, after which all measurements were repeated.

**Figure 2 cancers-13-05462-f002:**
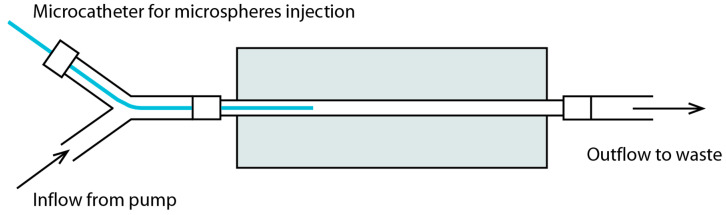
A schematic overview of the setup that was used to investigate the visibility of microspheres in a flow phantom. The grey box resembles the polyvinyl alcohol that is cast around the plastic tubing.

**Figure 3 cancers-13-05462-f003:**
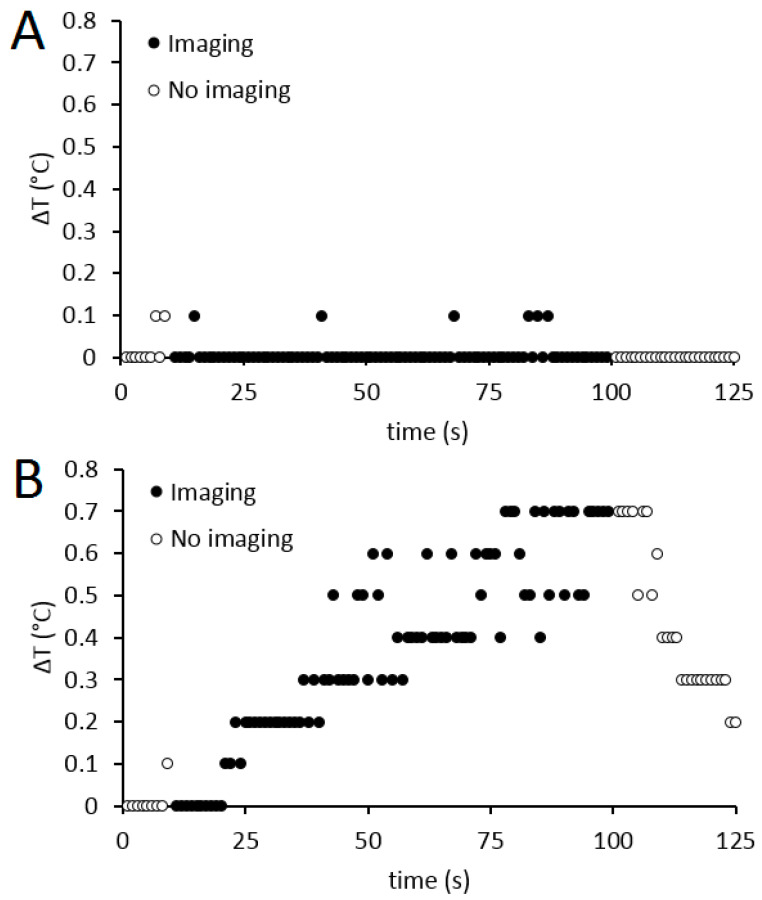
An example of the temperature curves as a result of radiofrequency-induced heating along the shaft of catheter A with microcatheter inserted (during which it was positioned extremely close to the bore). At t = 10, the T1 VIBE sequence (**A**) or the TRUFI RF_max_ sequence (**B**) was initiated and at t = 100, and the sequence was terminated.

**Figure 4 cancers-13-05462-f004:**
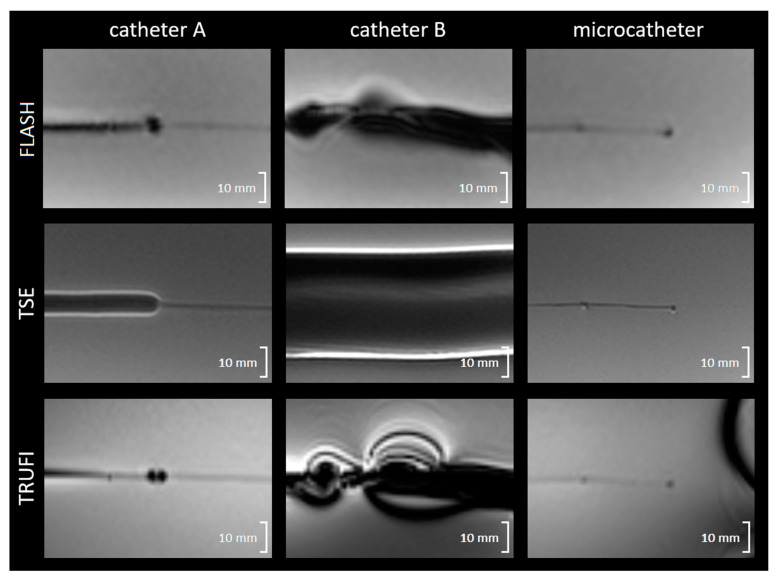
Visibility of three different catheters in an agarose phantom using different MRI sequences. FLASH = fast low-angle shot, TSE = turbo spin echo, TRUFI = true fast imaging with steady-state free precession.

**Figure 5 cancers-13-05462-f005:**
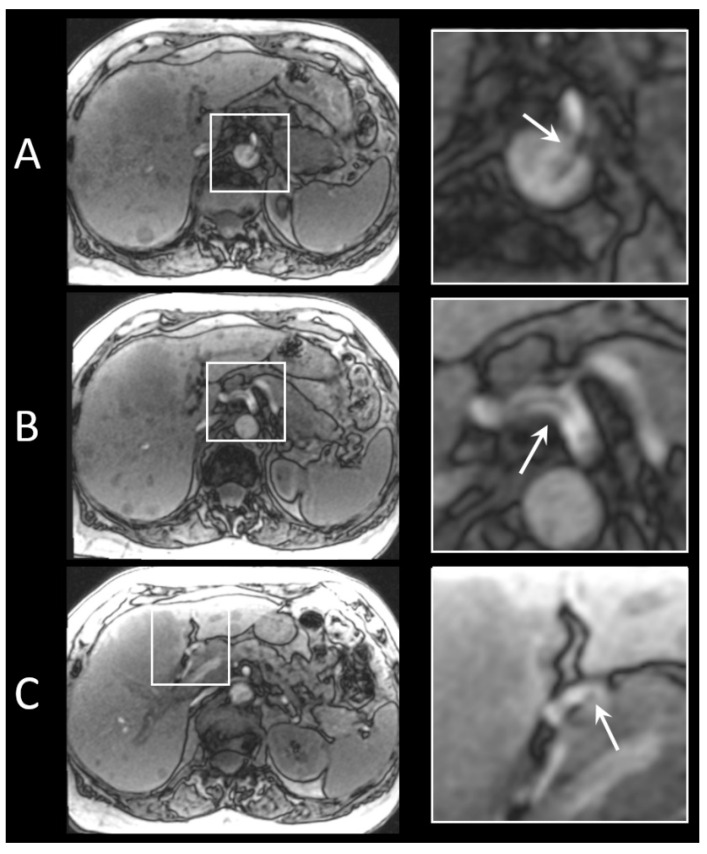
FLASH images of the two different catheters inserted in a patient treated with holmium SIRT for intrahepatic cholangiocarcinoma. (**A**) Catheter A exiting the aorta, entering the celiac trunk. (**B**) Catheter A entering the common hepatic artery. (**C**) Platinum/iridium marker at the tip of the microcatheter, positioned in the right hepatic artery.

**Figure 6 cancers-13-05462-f006:**
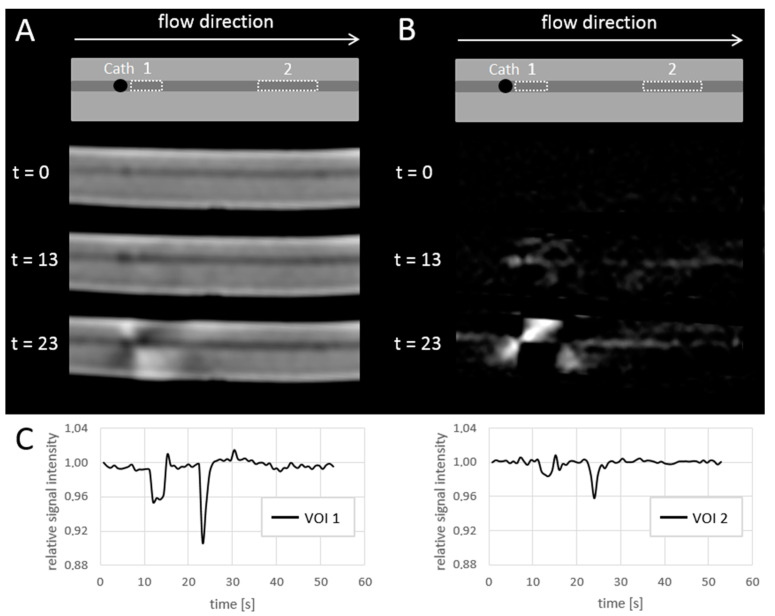
Intensity measurements over time in the flow phantom, during the injection of 100 mg/mL holmium-165 microspheres with flow set to 50 mL/min. (**A**) A schematic overview (top) of the images acquired, C = catheter tip, dotted lines indicate 2 VOIs used for signal quantification. Underneath, acquired images are shown at 3 different time points. (**B**) Subtraction images corresponding to the different time points of A, which more clearly visualize the signal loss induced by the microspheres. (**C**) illustrates the relative mean intensity per VOI over time. The first drop in intensity (t = 13 s) is a spill of microspheres after loading the catheter, and the second drop in intensity (t = 23 s) is flushing of the catheter with saline.

**Figure 7 cancers-13-05462-f007:**
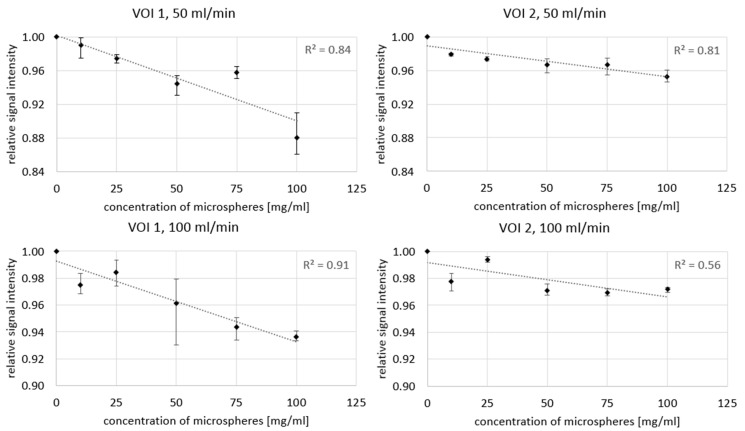
The lowest intensity value found in each of the intensity curves per injected concentration of holmium-165 microspheres (mean, error bars represent total range).

**Figure 8 cancers-13-05462-f008:**
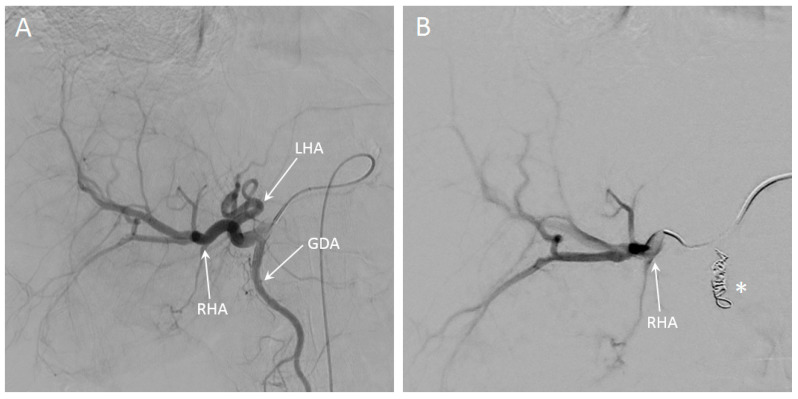
Overview angiography of the entire liver of a patient with intrahepatic cholangiocarcinoma (**A**), in which the left hepatic artery (LHA), right hepatic artery (RHA) and gastroduodenal artery (GDA) are visible. In (**B**), the RHA is selectively catheterized, after the GDA has been coiled (indicated with the asterisk).

**Figure 9 cancers-13-05462-f009:**
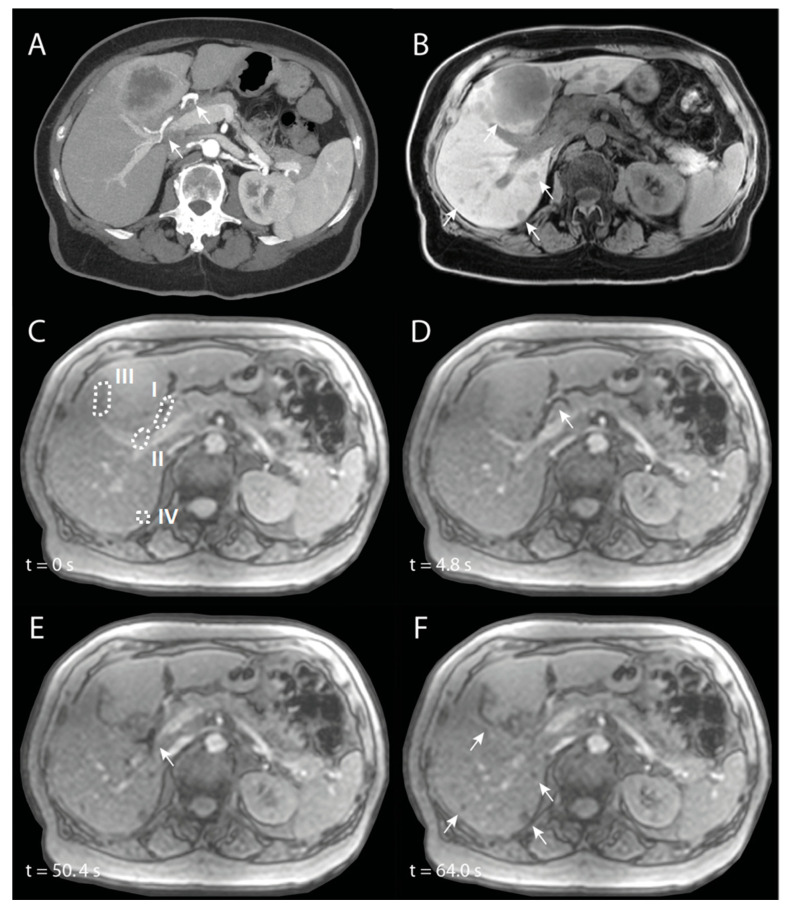
Imaging of a patient with intrahepatic cholangiocarcinoma, of whom the right hemiliver is treated with selective internal radiation therapy. (**A**) is a maximum-intensity projection of an arterial phase CT in which the right hepatic artery (RHA) is annotated with arrowheads. (**B**) is a corresponding non-enhanced T1-weighted MRI in which multiple tumours are visible (arrowheads). (**C**) to (**F**) are chronologic frames from the near real-time imaged holmium-166 microspheres injection: (**C**) is just before injection, with VOIs used for signal quantification annotated with dashed lines. In (**D**), there is loss of signal because of the microspheres in the proximal RHA (arrowhead) and in (**E**) in the more distal RHA (arrowhead). (**F**) is the end of near real-time administration, in which there is loss of signal mainly at the tumour sites (arrowheads) that was not visible prior to injection (**C**).

**Figure 10 cancers-13-05462-f010:**
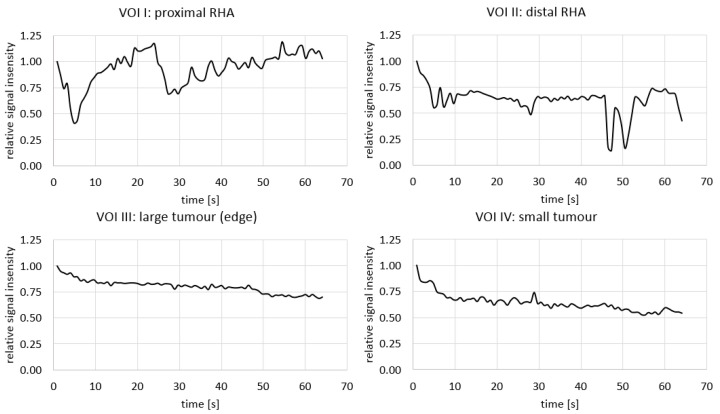
Average of the voxel intensity during near real-time imaging of holmium-166 SIRT in 4 different VOIs: proximal in the right hepatic artery (RHA; corresponding to the arrowhead in Figure 9D), more distal in the RHA (arrowhead in Figure 9E), the edge of the large tumour in segment 4B (top arrowhead in Figure 9F) and one of the smaller tumours (lowest arrowhead in Figure 9F). See Figure 9C for the delineation of the mentioned VOIs.

**Table 1 cancers-13-05462-t001:** Overview of the investigated sequence parameters for MR imaging.

Name	Sequence Type	TE/TR (ms)	Flip Angle (°)	Total Slices (*n*)	Slice Thickness (mm)	Field of View (mm × mm)	Matrix Size	Acquisition Time (min:sec)	Time-Averaged RF-Power (W)	Whole-Body SAR (W/kg)
T1 VIBE	Spoiled gradient echo	1.91/4.81	6	26	2.5	300 × 300	192 × 192	1:30	20.6	0.26
T2 TSE	Turbo spin echo	110/4490	150	31	3.0	160 × 160	256 × 230	1:30	113.2	1.52
T2 HASTE	Half fourier single-shot turbo spin echo	102/3000	180	29	5.0	266 × 266	256 × 256	1:30	151.5	2.00
TRUFI	Balanced steady state free precession	2.13/4.26	64	16	3.0	269 × 278	256 × 248	1:30	145.4	1.92
TRUFI RF_max_	Balanced steady state free precession	2.13/4.26	40	16	3.0	280 × 280	256 × 256	1:30	146.7	1.96

TE/TR = echo time/repetition time, RF = radiofrequency, SAR = specific absorption rate, VIBE = volumetric interpolated breath-hold examination, TSE = turbo spin echo, HASTE = half Fourier single-shot turbo spin echo, TRUFI = true fast imaging with steady-state free precession, TRUFI RF_max_ = worst case condition energy deposition.

**Table 2 cancers-13-05462-t002:** Radiofrequency-induced heating of the shaft of catheter A in two orientations: without microcatheter inserted (mc−) (Figure 1A), and with microcatheter inserted (mc+), during which it was positioned extremely close to the bore (Figure 1B). Range of maximum temperature differences is presented within brackets.

Sequence	Insertion Depth: 70 cm		Insertion Depth: 65 cm		Insertion Depth: 60 cm	
	∆T (°C), mc−	∆T 9(°C), mc+	∆T (°C), mc−	∆T (°C), mc+	∆T (°C), mc−	∆T (°C), mc+
T1 VIBE	0.00 (−0.1–0.1)	0.03 (0.0–0.1)	0.00 (0.0–0.0)	0.00 (−0.1–0.1)	0.03 (−0.2–0.3)	0.00 (−0.3–0.2)
T2 TSE	0.00 (−0.1–0.1)	0.17 (0.0–0.3)	0.00 (0.0–0.0)	0.13 (0.1–0.2)	−0.03 (−0.2–0.1)	0.13 (0.0–0.3)
T2 HASTE	0.00 (−0.1–0.1)	−0.03 (−0.1–0.1)	0.00 (0.0–0.0)	0.13 (0.1–0.2)	−0.03 (−0.1–0.1)	0.10 (0.0–0.2)
TRUFI	0.00 (−0.1–0.1)	0.37 (0.3–0.5)	0.03 (0.0–0.1)	0.27 (0.2–0.4)	0.00 (0.0–0.0)	0.40 (0.4–0.4)
TRUFI RF_max_	0.03 (0.0–0.1)	0.33 (0.3–0.4)	0.03 (0.0–0.1)	0.67 (0.6–0.7)	−0.03 (−0.1–0.1)	0.50 (0.5–0.5)

**Table 3 cancers-13-05462-t003:** Maximum artefact diameter induced by the different catheters in millimetres. The two values reported for catheter A represent the majority of the catheter length (left), and the thinner, distal 10 cm of the catheter (right). The two values reported for the microcatheter represent the shaft (left) and the marker at the tip (right).

Sequence	Catheter A	Catheter B	Microcatheter
FLASH	5.5/3.1	21.9	2.3/4.7
TSE	8.1/1.9	36.9	1.6/2.8
TRUFI	5.9/2.4	30.9	2.4/2.4

FLASH = fast low-angle shot, TSE = turbo spin echo, TRUFI = true fast imaging with steady-state free precession.

## Data Availability

The data presented in this study are available upon reasonable request to the corresponding author.

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
