# Peer review of "Development of an MRI-Guided Approach to Selective Internal Radiation Therapy Using Holmium-166 Microspheres"

_cancers, 2021, doi:10.3390/cancers13215462_

Round 1
Reviewer 1 Report
All of my points have been adequately considered and the manuscript has improved significantly.
Author Response
We want to thank the reviewer again for their time and valuable feedback, as it has definitely improved the manuscript.
Reviewer 2 Report
Thank you for adressing my comments, of which most have been satisfactorily answered.
Your explanations about the control of the catheter position in the MRI could not invalidate my objections, because still a possible secondary dislocation of the catheter can remain unnoticed, which can, with a very high probability, be fatal during an application of the radiopharmacon .
For real-time imaging during injection, even after the catheter location has been verified, a volumetric sequence is necessary. The study should mention this, if not even should be repeated with this parameter changed.
Author Response
Thank you for addressing my comments, of which most have been satisfactorily answered.
Your explanations about the control of the catheter position in the MRI could not invalidate my objections, because still a possible secondary dislocation of the catheter can remain unnoticed, which can, with a very high probability, be fatal during an application of the radiopharmacon.
For real-time imaging during injection, even after the catheter location has been verified, a volumetric sequence is necessary. The study should mention this, if not even should be repeated with this parameter changed.
First, we want to thank the reviewer again for their time and valuable feedback, as it has definitely improved the manuscript.
We do agree with the reviewer that using a volumetric sequence for real-time imaging during injection would be very beneficial for the procedure. We consider it an important aspect for future research to further improve our MRI-guided approach. Implementation of a volumetric real-time sequence would be a research objective on its own, as it is not straight forward to implement and there is of course a trade-off between temporal resolution, image resolution and quality, et cetera.
The stability of a catheter position is very dependent on the individual patient anatomy and during the clinical study, we considered the odds of a secondary dislocation as described rather low. Additionally, in an image-guided procedure, the dose distribution in the entire liver volume would be evaluated in between injections of the microspheres, and it would therefore be noticed if (part of) the dose ends up in a different part of the liver. This would of course still be unfavorable.
All in all, we do agree that a volumetric sequence would be beneficial. Therefore, we have added the following passage to the discussion (lines 430-435):
“Moreover, an MRI-guided procedure would greatly benefit from continuous 3D imaging during the injection. This could resolve the issue of the vessel moving out of plane and would further increase the safety of the procedure as it allows for continuous visualization of the microsphere injection and catheter position, but further work is needed to investigate the feasibility and implementation of such acquisition methods.”
We hope that this meets the reviewers expectations.

Reviewer 3 Report
This is a revised version of a previously submitted manuscript. The authors have considered all comments raised by this reviewer and the paper has improved both in terms of scientific content and logical flow.
Author Response

(The authors gave the same response as above.)

Round 2
Reviewer 2 Report
Dear Authors,
Unlike the other reviewers, I consider the conclusions on the safety of this procedure based on results obtained in a single (!) patient not justified in any way, and inappropriate especially for the professional audience (which is limited to studying the abstract and information easily accessible in databases).
The main point of the article is the increased patient safety during radioembolization by live monitoring of the applied radioactivity by measuring the accumulation of the surrogate parameter holmium microspheres. Secondary catheter dislocations are only briefly discussed in the text as a disadvantage. But the clinical experience of any interventionalist performing such procedures will show that most essential for safe performance of such a procedure is the reliable monitoring of the catheter course in the hepatic artery. Individually not pre-quantifiable tension conditions in the catheter can, in the worst case, lead to dislocations, resulting not only in erroneous embolization into other liver areas, but also into the flow area of the gastroduodenal artery, etc.
I do not consider promoting a method as "safe" to be justified without the possibility to simply monitor the exact catheter course, including catheter tip. The phantom part of this study is quite well done, however, for the reasons just stated, I do not think mixing it with a single patient case (perhaps one that went well by chance) is legitimate.
As you write, 3D monitoring in the patient is worth its own study, which will certainly be a worthwhile one, because it is indispensable to allow such a procedure to be considered “safe for the patient”. The use of a single case description (without monitoring the exact catheter course) as a supplement for a phantom fundamental research paper does not seem permissible, or in any way sufficient basis for a general conclusion on patient safety.
Publication of the contents of this article can only succeed if the phantom part—which certainly has reason to be published—is considered separately from the patient case part. This avoids drawing a too guileless picture of radioembolization.
Author Response
Reviewer #2
Dear Authors,
Unlike the other reviewers, I consider the conclusions on the safety of this procedure based on results obtained in a single (!) patient not justified in any way, and inappropriate especially for the professional audience (which is limited to studying the abstract and information easily accessible in databases).
The main point of the article is the increased patient safety during radioembolization by live monitoring of the applied radioactivity by measuring the accumulation of the surrogate parameter holmium microspheres. Secondary catheter dislocations are only briefly discussed in the text as a disadvantage. But the clinical experience of any interventionalist performing such procedures will show that most essential for safe performance of such a procedure is the reliable
monitoring of the catheter course in the hepatic artery. Individually not pre-quantifiable tension conditions in the catheter can, in the worst case, lead to dislocations, resulting not only in erroneous embolization into other liver areas, but also into the flow area of the gastroduodenal artery, etc.
I do not consider promoting a method as "safe" to be justified without the possibility to simply monitor the exact catheter course, including catheter tip. The phantom part of this study is quite well done, however, for the reasons just stated, I do not think mixing it with a single patient case (perhaps one that went well by chance) is legitimate.
As you write, 3D monitoring in the patient is worth its own study, which will certainly be a worthwhile one, because it is indispensable to allow such a procedure to be considered “safe for the patient”. The use of a single case description (without monitoring the exact catheter course) as a supplement for a phantom fundamental research paper does not seem permissible, or in any
way sufficient basis for a general conclusion on patient safety.
Publication of the contents of this article can only succeed if the phantom part—which certainly has reason to be published—is considered separately from the patient case part. This avoids drawing a too guileless picture of radioembolization.
Dear reviewer,
Thank you for your extensive review and expression of your concern. First of all, we understand and share your concerns regarding patient safety and would like to stress that patient safety has always had our highest priority in the development of any clinical trial to study novel treatment methods. In fact, the current paper describes the safety aspects that were addressed prior to
conducting a phase I clinical trial. Only after this extensive safety testing, we designed and conducted a small pilot study (N=6) of the developed MRI-guided approach designed (NCT: https://clinicaltrials.gov/ct2/show/NCT04269499), which has feasibility and clinical safety as the main outcome parameters. In addition to the mentioned safety precautions resulting from the present manuscript, additional fallback procedures are in place to prevent injection of activity to undesired locations and detect catheter dislocation. The full methodology and results of this pilot study will be published separately once follow-up is completed.
In the presented article, our main goal was to demonstrate the results of this safety testing, investigating the MRI-safety of the used catheters in an ex vivo setting and investigating the visibility of holmium microspheres in a flow phantom during injection. We have opted to provide an exemplary initial patient case originating from the subsequent clinical trial as to illustrate our
preliminary in vivo findings with regards to catheter and holmium visibility on MRI. We apologize if the suggestion was raised that this initial patient case was meant to claim procedural safety, which should be established in a larger patient group and is one of the main outcome parameters of the present pilot study.
We have made changes to the manuscript to better reflect this distinction and thank the reviewer for their useful critiques to improve the message of the manuscript, please see:
- simple summary and abstract, lines 20 and 34-38
- introduction, lines 97-101
- discussion, lines 367-370
- conclusion, lines 462-472
We hope that the reviewer agrees with us that in this context the patient case has value as an example of actual in vivo imaging during an MRI-guided SIRT procedure. We fully agree with the reviewer on the potential improvements that could be made in future work (regarding the 3Dimaging during injection and constant visualization of the catheter tip), as also stated in our previous reply, and which have been pointed out in the discussion.
To summarize, we hope that we have made the distinction between our ex vivo experiments and the in vivo comparison more clear, and have improved the clarity regarding our statements on the safety of the MRI-guided procedure. We would prefer not to omit the clinical example from the article, as it does support the non-safety related ex vivo findings and hope that the review agrees
with this.
We would like to thank the reviewer again for their consideration.

This manuscript is a resubmission of an earlier submission. The following is a list of the peer review reports and author responses from that submission.
Round 1
Reviewer 1 Report
The purpose of this study was to investigate the MR safety of three conventional angiography catheters for use during MRI-guided SIRT at 3 T and to investigate the correlation between 166 Ho microsphere concentration and detectability in near realtime MRI in a flow phantom including an initial patient case.
The paper is well written and sound.
A few details have to be annotated in the following regarding the presentation:
198 Please include here as well the information that inclusion/exclusion criteria are available at https://clinicaltrials.gov/ct2/show/NCT04269499
207 'near real-time imaging' -> please give temporal resolution
210/211 'as long as possible' -> how long was this approximately? 5s, 10s, .. 30s, ... longer?
213-220 Regarding VOIs please include reference to Figure 6 and 8.
248/249 What was the maximal force the catheter experienced?
296 Figure 6: Using the same label 'C' for the catheter and sub-figure 'C' may be confusing in the Figure legend. You may use 'Cath' or something else as label for the catheter.
302 Figure 7: Increase font size of all figure labeling.
312 Video: It would be convenient for the reader if the video could be provided with arrows pointing to the relevant spots like in Figure 8F and if the respective frame were injection(s) took place would be labeled as well.
322/339 Individual labeling of the four VOI's (e.g. I-IV or a-d) in Fig 8C/8F as well as in Fig.9 would make it easier for the reader.
335 Fig.9 suggest this should be '... until around 30 seconds...' instead of 40, isn't it?
Reviewer 2 Report
Development of an MRI-guided approach to selective internal radiation therapy using holmium-166 microspheres
The authors present the model of selective internal radiotherapy (SIRT) based on holmium microspheres combined with real-time monitoring of the microspheres with MRI. In this context, they present the potential MR suitability of two catheters and a superselective catheter in an in vitro model. Furthermore, in vitro measurements of the passage of holmium microspheres were performed in a perfusion model. The principal feasibility is finally demonstrated using a single patient case.
Real-time monitoring of the distribution of SIRT microspheres is a relevant clinical issue. In principle, this is possible with MRI. However, the question I have is how exactly the authors envision proceeding if one notices an inadequate distribution of SIRT-particles on MRI and wants to reposition the catheter (angiographically). In my opinion, the study experiences a major limitation in that only a single MRI slice was used for positional assessment. Discussing reasons for this and pointing out possible alternatives would be essential points that would still need to be discussed in the discussion section. Also, it seems very dangerous to me not to be able to check the catheter position immediately angiographically, since one must bring the patients from the MRI to the angiography suite first in a time-consuming way.
Personally, based on the in vitro testing presented, I would not dare to put a patient with a catheter into an MRI. With two catheters, one moves when you get too close to the magnet, the possible reasons and implications are not discussed in the text. Finally, as the authors correctly mention, cone-beam CT or the combination of a ring CT with an angiography unit is also a technique to measure real-live control of particle distribution in SIRT. MRI, in my opinion, has now too many limitations compared to these alternatives to be promising in clinical practice.
Abstract:
OK
Introduction:
OK
Materials and Methods:
* Please specify the ASTM standard test methods since this might be unknown to most of the readers.
Results:
OK
Discussion:
Please discuss the proceeding if one notices an inadequate distribution of SIRT-particles on MRI and wants to reposition the catheter (angiographically). Please discuss the limitations of only measuring one MR slice and compare this to the alternatives cone-beam CT and conventional CT.
Conclusions:
The summary is too optimistic. It does not consider that only one MRI slice was used, and thus no reliable statement can be made about the situation in the entire liver and possibly in collateral vessels opening during the intervention. Furthermore, the MRI capability in these few measurements may also be just coincidence. It should be formulated a little more cautiously.
Table .:
OK
Figures:
1) Why is the catheter bent in 1B)? Please explain in detail.
8) The presentation of overview angiography of the entire liver and superselective angiography of the right hepatic artery as a comparison would improve the comprehensibility of the images.
References:
OK
Reviewer 3 Report
General comments:
The paper authored by Roosen et al presents the preliminary results – both in phantom and in vivo – of an MRI-guided technique for internal radiotherapy using holmium-166 microspheres in hepatic cancer, with the aim of a more personalised approach to therapy.
Radioembolization via beta emitters, particularly with the more novel holmium-166, for hepatic malignancies has gained popularity over the last decade; nevertheless, there are still challenges regarding image guidance and control of radiation delivery.
The manuscript is well written and comprehensive. I only have some minor suggestions to the authors to consider:
The Introduction would benefit from a paragraph succinctly describing the physical and radiobiological properties of the currently used radioisotopes for internal targeted therapy in liver cancer. Justify the use of beta emitters through their energy / dose rate, half-life and range in tissue. Furthermore, add 1-2 sentences on the radiobiological advantage of targeted therapies for cancers (such as liver) that are difficult to treat via conventional chemo-radiotherapy techniques.
Also, justify your choice of using Ho-166 instead of Y-90 using physical / radiobiological principles and also the advantages regarding imaging modalities resulting from holmium’s physical properties.
In the Discussion mention, among limitations, that this was a case study (1 patient-only study) and the acquired imaging information / outcome might also be patient-dependent, which would justify larger accrual.
Specific comments:
- Line 71 – use the usual referencing system for clinicaltrials.gov, i.e., allocate a ref. number and add the reference to the end of the manuscript as a url
- Line 102 – ‘…following the American…’
- Lines 109-113 – ‘In both guiding….’ - divide the phrase into two sentences to ease the understanding
- Line 118 – quantify ‘even closer’